



# Response of tropical marine benthic diatoms exposed to elevated irradiance and temperature

Sazlina Salleh[1,2], Andrew McMinn[3,4]

[1]Centre for Policy Research and International Studies (CenPRIS), Universiti Sains Malaysia, 11800, Minden, Pulau Pinang, Malaysia

[2] Centre for Marine and Coastal Studies (CEMACS), Universiti Sains Malaysia, 11800, Minden, Pulau Pinang, Malaysia

[3]Institute of Marine and Antarctic Studies (IMAS), University of Tasmania, Box 252-77, Hobart 7001, Tasmania, Australia

[4]College of Marine Life Science, Frontiers Science Center for Deep Ocean Multispheres and Earth System, Ocean University of China, Qingdao 266003, China

*Correspondence to*: Sazlina Salleh (sazlina@usm.my)

**Abstract.** Shallow tropical marine environments are likely to experience future water temperatures that will challenge the ability of life to survive. Changes in temperature and irradiance during tidal cycles in the Tanjung Rhu estuary, Langkawi, Malaysia in 2007 did not significantly affect the benthic diatom communities, although, higher photosynthetic parameters, such as maximum relative electron transport rate (rETRmax), photosynthetic efficiency ($\alpha$), maximum quantum yield ($F_v/F_m$) and effective quantum yield ($\Delta F/F_{m'}$), were recorded at high tide when the temperatures were lower. However, when benthic diatoms were experimentally exposed to irradiances of 1800 µmol photons $m^{-2}$ $s^{-1}$, they were only able to photosynthesize at temperatures < 50 ºC. Above this temperature, no photosynthetic activity was observed. Not only did high temperatures at high irradiance affect the algal communities, but limited photosynthetic activity was also observed in samples when exposed to limited irradiance. Recovery rates were highest at the lowest temperatures and decreased as the temperature increased. The recovery rates for samples exposed to temperatures of 40 ºC were 4.01E-03 ± 0.002 and decreased to 1.01E-05 ± 0.001 at 60 ºC, indicating irreversible damage to Photosystem II (PSII). These characteristics suggest that the benthic diatom communities in this estuary are already suffering from thermal damage and that enhanced photoinhibition would result if exposed to elevated temperatures, especially during low tide. 50 °C is apparently a temperature threshold for tropical estuarine benthic microalgal communities. Future warming is likely to cause this temperature to occur more frequently, which will cause a reduction in benthic primary production.

## 1 Introduction

Benthic diatoms make a significant contribution to the primary production of shallow coastal and estuarine environments, regularly contributing more than 50% of the total annual production (Underwood 2002, Rajesh et al. 2001). A few key environmental factors influence biomass production. For instance, high turbidity, which is a characteristic of many tropical



estuaries, reduces the irradiance reaching the microalgae mats which then limits photosynthesis (Underwood 2002). Intertidal ecosystems in the tropics are affected by large changes in solar irradiance and temperatures, caused by tidal cycles. During prolonged low tides, variation in irradiances, temperatures, and salinity can adversely affect the photosynthetic activity benthic microalgal communities (Underwood 2002, Laviale et al. 2016). To overcome these problems, they have developed adjustment and avoidance strategies. They can protect themselves physiologically against the damaging effects of exposure (Blanchard et al. 2004, Perkins et al. 2010) by dissipating the excess energy absorbed by light harvesting centers, a process known as non-photochemical quenching (NPQ), or they can move vertically in the sediment, adjusting their position to minimize irradiance (Consalvey et al. 2004a, Perkins et al. 2010, Cartaxana et al. 2013). For instance, in a study by Mouget et al. (2008) it was shown that benthic diatoms with a high migratory capacity could experience high photosynthetic activity without any evidence of photo damage, while non migratory taxa showed a significant depression in photosynthetic activity, which was caused by high irradiance and/or UV-B.

It is widely known that temperature plays an important role in the growth rate of all algal cells (Eppley 1972, Davison 1991, Eggert and Wiencke, 2000, Longhi et al., 2003). Prolonged exposure to temperatures significantly above ambient (ideal growth temperature) also leads to a decrease in the concentration of chlorophyll *a* per cell (Defew et al., 2004). Excess light energy can also subsequently cause a decline in photosynthetic activity, which is reflected in the response of Rapid Light Curve (RLC) parameters (Du et al., 2018). With combined stresses, such as high temperature and irradiance and/or UV-B), damage to Photosystem II can be induced, impacting the recovery process; this is apparently more significant in planktonic diatoms by comparison with benthic species (Wu et al., 2017). Photoinhibited cells can mostly regain their capacity for photosynthesis when removed from a high light environment (Wu et al., 2017). The ability of algae to tolerate dynamic intertidal environments such as light and temperature changes during tidal cycles and periods of photoinhibition largely determines estuarine community species composition (Derks and Bruce, 2018). If photoprotective mechanisms are inadequate to mitigate excess light, overexcitation of PSII can occur leading to the production of reactive oxygen species (ROS), thus damaging the photosynthetic apparatus (Müller et al., 2001). The extent of photooxidative damage and the speed of recovery are related to the degradation rate of D1 protein and the subsequent re-synthesis and replacement by new proteins (Derks and Bruce, 2018; Domingues et al., 2012). Similarly, benthic diatoms inhabiting sediment surfaces in estuaries are usually able to recover from extreme temperatures and light photoinhibition caused by tidal changes, since photoinhibition is only rarely recorded in these communities (Defew et al., 2004; Lee and McMinn, 2013; Perkins et al., 2006). Furthermore, many have the ability to migrate vertically, hence positioning themselves within the sediment at a depth that provides shading from excess irradiance and provides an optimal light environment for their photosynthetic activities (Cartaxana et al., 2011; Mouget et al., 2008; Perkins et al., 2010).

Due to the major contribution of benthic diatoms to ecosystem processes, they have been widely studied in tropical coral reefs and intertidal flats and coastal intertidal zones (Mitbavkar and Anil 2002; Mitbavkar and Anil 2004; McMinn et al.,





2005; Underwood, 2002). However, there are relatively few studies of benthic diatoms in tropical estuaries (Patil and Anil 2008). Although the photosynthesis of tropical benthic diatoms is likely to be highly sensitive to elevated temperature, comparatively little attention has been given to the combined effects of both elevated light and temperature. Water temperatures in tropical marine ecosystems are already high and relatively small increases can have severe negative impacts. Intertidal and subtidal communities are particularly vulnerable and increasing temperatures, resulting from global warming, will have unpredictable consequences. This study examines the impact of high light and temperature as temperatures increase. These conditions are predicted to worsen with the increase in seawater temperatures resulting from climate change. The aim of this study is to determine the stress response of diatom physiology to the combined effects of elevated irradiance and temperature in diatom dominated micrphytobenthos (MPB) assemblages and understand their subsequent recovery.

## 2 Materials and methods

### 2.1 Study area and field sampling

Tanjung Rhu is located on the north coast of Langkawi, Malaysia (Fig. 1). The area has a humid, tropical climate with daily maximum temperatures between 27.0 and 40.0ºC. This estuary contains a range of habitats (beaches, mangroves, and seagrass beds) all within close proximity. The estuary is mesotidal with semi-diurnal tides with a tidal range between 2 m and 3 m. Much of the ecosystem is exposed during low tide and submerged during high tide. The estuary is surrounded by a mangrove forest, mainly dominated by *Rhizophora* and *Avicennia* species. The benthic habitat is mostly composed of coarse sand with patches of mud. The water normally carries a high-suspended load, originating partly from the mangrove forest and from the river sediment itself.

To gain a better understanding of the surrounding environment, samples were collected in April 2007 from three sites (Site A, B and C) all in proximity (20 meters) to the intertidal zone of the Tanjung Rhu Estuary, during both high and low tide. Description of sites A, B and C are provided in Table 1. Environmental data (salinity, temperature) and photosynthetic parameters were collected at ebb and flood tide. Water height at ebb tide was approximately 0 to 0.2 m and 0.5 to 1 m at flood tide. On each occasion, seven 15 mm diameter hand-pushed sediment cores were taken; three for photosynthetic parameter analysis, three for chlorophyll *a* analysis and one for species composition. For the temperature incubation experiments, the top 10 mm (approximately) of the sediment was manually scraped off and placed in a dark plastic bag (20 x 20 cm). Samples were stored in the dark and promptly returned (approximately 15 min) to the laboratory for the experiments.

At each sampling site, water quality measurements (temperature, salinity, and photosynthetically active irradiance (PAR)) were measured using a Hydrolab Datasonde 4a (Hach, Loveland CO). Water samples for all parameters and nutrient analyses were collected during high tide (water level: 1.0 m) and low tide (water level: 0.2 m). Nitrate, phosphate, and ammonium were analyzed on a Hach Kit DR 2000 Spectrophotometer (Hach, Loveland CO). Ammonia, expressed as ammonia-nitrogen ($NH_3$-N), was determined using the Nessler method (Hach Company, 1995). Phosphate, expressed as





phosphorus-phosphate ($PO_4^{3-}$-P) was determined by the PhosVer (ascorbic acid) method (Hach Company, 1995). Nitrate, expressed as nitrate-nitrogen ($NO_3^-$ -N) was determined by the cadmium reduction method (Hach Company, 1995).

## 2.2 Community composition

For species composition, the top approximately 5 mm of sediment was collected using a 15 mm diameter hand-pushed mini corer during low tide. The samples for were transferred into a 15 mL vial and fixed with 2% glutaraldehyde solution prior to processing. Organic material was removed by digestion in a 10% hydrogen peroxide solution for 72 hours. The sample processing methods for diatoms identification followed Cunningham et al. (2003). The species composition was assessed using a Zeiss S20 Microscope (Zeiss, Germany). A minimum of 400 individuals was counted per sample. The diatoms

species were expressed as a relative abundance (% total microalgae) of frustules counted. Due to limited taxonomic information on tropical benthic diatoms, identification was mostly based on appropriate references from the temperate areas (Beltrones and Castrejon, 1999; Cooper, 1995; Round et al., 1990; Stidolph, 1980; Tomas, 1997; Witkowski, 2000). Only empty frustules of benthic diatom were studied, so it was not possible to establish which species were actively photosynthesizing or growing in the sediment at the time of collection.

## 2.3 Chlorophyll *a* biomass


Chlorophyll *a* was measured as a proxy for algal biomass. The chlorophyll biomass was determined following (Jordan et al., 2010). For chlorophyll *a* biomass and fluorescence analysis, a 45 mm diameter clear polycarbonate cores were manually pushed into the sediment and stoppered using a rubber bung and immediately returned to temporarily established working place. Three cores were taken for three for biomass analysis and three fluorescence analysis (see PAM Chlorophyll

Fluorescence measurements). The sediment for chlorophyll *a* analysis, for both low and high tide samples, were placed in an ice-filled, light-proof container and immediately transferred to the laboratory. The top 10 mm of sediment was re-suspended in 10 mL methanol, thoroughly mixed and then stored in the dark for 12 hours at 4 ºC. After the sediment had settled, the solvent was decanted off to measure the chlorophyll *a* content using acidification methods (Holm-Hansen and Lorenzen, 1965). A Turner Designs 10AU Fluorometer (Sunnyvale, CA, USA) was used to measure the chlorophyll *a*. The fluorometer

was calibrated against a chlorophyll *a* standard (Sigma Chemical Co., St. Louis, MO, USA).

## 2.4 PAM Chlorophyll Fluorescence measurements

Variable fluorescence was measured using a Pulse Amplitude Modulation (PAM) (Schreiber 2004) fluorometer comprising a computer-operated PAM Control Unit (Walz, Effeltrich, Germany) and a WATER-EDF-Universal emitter-detector unit

(Gademann Instruments GmbH, Wurzburg, Germany). The top 10 mm of each core was sectioned on site, 5 for PAM fluorometry analysis. Each sample was mixed with filtered seawater and transferred into a vial wrapped in aluminium foil to protect them from light and to dark-adapt the samples for 15 min. The samples were shaken vigorously and then allowed to





settle for approximately 10 s before analysis. The PAM methodology followed (McMinn et al., 2005) and Salleh and McMinn (2011).

To calculate the maximum PSII quantum yield ($F_v/F_m$) the samples were dark adapted in the field for 15 minutes by wrapping the jars in foil and placing them in a dark container immediately after sampling. Rapid light curves (RLCs) were taken under software control (Wincontrol, Walz) to obtain values for $F_v/F_m$, NPQ, and $E_k$ (Ralph and Gademann, 2005). Red light emitting diodes (LED) provided the actinic light used in the RLCs at levels of 0, 85, 125, 194, 289, 413, 517, 1046 and 1554 µmol photons m$^{-2}$ s$^{-1}$. The saturation pulse irradiance was 3000 µmol photons m$^{-2}$ s$^{-1}$ for a period of 0.8 seconds.

Samples were exposed to each light level for 10 seconds. Photomultiplier gain settings were between 3 and 8 (Wincontrol, Walz). $E_k$, the light saturation parameter, was calculated from the intercept between the maximum relative electron transport rate (rETR) and α, the photosynthetic efficiency (Falkowski and Raven 2007). The rETR was calculated by multiplying the irradiance by the quantum yield measured at the end of that interval. PAR versus rETR curves were described using the model of Jassby and Platt (1976) using multiple non-linear regression curve fitting protocols on SPSS software (SPSS Inc.,

Chicago, IL, USA).

NPQ is a measurement of the activity of the protective mechanism, which is designed to protect against over reduction of the photosynthetic electron transport chain by dissipation of excess absorbed light energy in the PSII antenna system as heat (Ruban et al., 2004). NPQ is a complex response comprising at least four different, time dependant, responses Fanesi et al. (2016). It is recognised that it is not possible to distinguish process is occurring using this approach. NPQ was determined by

the following equation (Schreiber, 2004).

$$NPQ = (F_m - F_{m'}) / F_{m'} \qquad (1)$$

Non-photosynthetic quenching (NPQ) values presented here were taken from the RLCs data generated by Wincontrol software (Walz GmbH, Effeltrich, Germany). The short light exposure time in this experiment (10 s RLCs duration), would have been insufficient for maximum NPQ development and so are only a relative measure of the ability of the treatments to

develop NPQ (Ralph and Gademann, 2005).

### 2.5 Temperature and irradiance experiments

For the temperature and irradiance incubation experiments samples were collected during low tide and exposed to a combination treatment of irradiances (1800, 890, and 0 µmol photons m$^{-2}$ s$^{-1}$ and high temperatures (30, 35, 40, 45, 50, 55

and 60 °C). Samples were always taken from Site C, where the highest temperatures were recorded during *in-situ* data collection (Table 1). The top 0.5 to 10 mm of the sediment was collected by hand using a clear polycarbonate core during low tide and wrapped in a dark plastic bag prior to analysis. Samples were kept upright in the tubes and were immediately returned to lab for the experiments. The sediment samples were suspended in a beaker and excess seawater decanted.



Filtered seawater collected from the same site was added to each vial and the vials were placed in three chambers of different
irradiance in a temperature-controlled water bath (Lauda 20 L, Brinkmann Lauda Inc. Konigshoven, Germany). Triplicate
sample vials were placed in each irradiance chamber. Fresh algal samples were used for each temperature change. Since
irradiance was provided by a halogen lamp (Thorn 300W, Borehamwood, UK), it is understood that the spectrum will have
differed from that of natural sunlight. The irradiance in each chamber was measured with a Quantum-Light Meter L1-189
(LI-COR, Nebraska, USA). The irradiance of each chamber was adjusted by the addition of several layers of dark plastic
filters. These filters also assisted in reducing the heat produced by the halogen lamps. Care was taken to ensure that the
temperature remained constant throughout the incubation. Fresh samples were used for each temperature incubation. The
algal samples were exposed to three different irradiance treatments A: 1800 μmol photons $m^{-2}$ $s^{-1}$, B: 890 μmol photons $m^{-2}$
$s^{-1}$, and C: 0 μmol photons $m^{-2}$ $s^{-1}$ with three replicates for each treatment. Samples were placed in the water bath for a period
of one hour for each temperature treatment. The temperatures selected were 30 °C, 35 °C, 40 °C, 45 °C, 50 °C, 55 °C and 60
°C. Lower temperatures (30 °C to 35 °C) were used as controls since the community had been growing within that
temperature range in the natural habitat. As experiments were conducted in a makeshift laboratory, temperatures <30°C
could not be obtained due to equipment limitations. At the end of each incubation, each replicate was immediately
transferred into a cuvette for photosynthetic analysis. A Water PAM fluorometer was used to determine the quantum yield at
the end of each temperature treatment. RLCs (see above) were run to obtain values for $F_v/F_m$, rETRmax, $E_k$ and α. In each
treatment, three replicates (n = 3) were measured. Before measurement, samples were gently shaken to ensure even mixing
and to limit settlement and were then placed inside the measuring cuvette of the Water PAM fluorometer. Measurements
were conducted as soon as possible to ensure thermal change minimized during the measurements of the RLCs.

**2.6 Rate of recovery**

The method of determining and analyzing the rate of recovery was adapted from McMinn and Hattori (2006), Salleh and
McMinn (2011). The recovery measurements were made after the one-hour light incubation. The recovery rates suggest the
ability of cells to reactivate photosynthetic activity after one-hour exposure to high light incubation. The measurement of
recovery from photoinhibition was made using the pre-installed routine RLC + Recovery of the Water-PAM. After the last
actinic light period of the RLC, the sample was left in the dark, while $F_v/F_m$ was determined after 30, 60, 180, 300 and 600 s.
The rate of recovery from photoinhibition was calculated following Oliver et al. (2003):

$$\Phi_t = \Phi_I + (\Phi_M - \Phi_I)(1 - e^{-rt})$$ (2)

Where $\Phi_t$ is the given value of $F_v/F_m$ at time t (s), $\Phi_I$ is the initial value of $F_v/F_m$ before recovery measurements have
commenced, $\Phi_M$ is the fully recovered value and r is an exponential rate constant for the recovery of $F_v/F_m$ from
photoinhibition ($s^1$). This formula assumes that the recovery rate of $F_v/F_m$ is dependent on the degree of damage.



### 2.7 Statistical analysis

The mean and standard deviations were calculated from three independent replicates of each parameter. To evaluate the effects of temperature and irradiances (fixed factor) on the photosynthetic parameters ($F_v/F_m$, rETRmax, $E_k$, α, NPQ and recovery rate), a two-way analysis of variance (ANOVA) and regression were used with subsequent Tukey *post hoc* comparisons. Differences were accepted as significant at *P < 0.05* unless otherwise stated.

### 3 Results

The diatom genera *Cocconeis, Navicula* and *Rhopalodia* dominated the benthic habitats, which predominantly consisted of course sand with mud/silt patches. The sediment characteristics were similar among the sampling sites. The most common species found were *Cocconeis placentula, Navicula raphoneis, Navicula clipeiformis* and *Rhapalodia acuminatea*. The composition of the diatom communities at all sites and tides were similar without any differences in dominant species. There were significant differences in environmental parameters between high and low tide (Table 2 and Table 4(a)), however, environmental characteristics between sites were similar except for bottom irradiance. Tidal exposure caused significant differences in the irradiance reaching the bottom and elevated the temperature at low tide for all sites. During high tide, bottom irradiances were at their lowest (173.42 ± 13.73 µmol photons m$^{-2}$ s$^{-1}$) at the shaded Site B and were highest in the area that was most exposed, Site C (1964.33 ± 99.68 µmol photons m$^{-2}$ s$^{-1}$), when the algal mats were almost fully exposed. There was a more than six-fold increase in the irradiance from low tide to high tide (Table 2 and Table 4(a)). Water temperature rose from 28 °C during high tide to 43 °C during low tide. The highest nutrient levels were recorded during high tide, when seawater re-entered the estuary (Table 2 and Table 4(a)) There was a least a 50% increase in nitrate and phosphate levels during high tide, although the changes were not significant (Table 2).

### 3.1 In situ chlorophyll *a* and photosynthetic parameters of benthic diatom

Due to the high turbidity in the Tanjung Rhu estuary, diatoms on the sediment surface were generally exposed to significantly lower irradiances at high tide than during low tide. Differences in sediment surface PAR caused a significant decline in chlorophyll *a* biomass and PSII activity, as measured in the RLCs photophysiological parameters. The highest biomass was observed at Site B (Shaded area) where PAR was at the lowest during the sampling time. At low tide the chlorophyll *a* values were between an average of 17.6 to 21.22 mg chl *a* m$^{-2}$, but values increased by almost 50% at high tide (20.19 to 37.38 mg chl *a* m$^{-2}$). The photosynthetic parameters for both low and high tide are presented in Table 3. Quantum yield, rETRmax and $E_k$ were relatively low during both high and low tide. However, the effects of the tidal cycle could be clearly seen, as most values increased as the tide rose, except for the $E_k$. The $E_k$ values declined 23% during high tide. The NPQ values were higher at low tide than at high tide. In summary, differences in all parameters were significant between tides by comparison to sites (Table 4(b)).



220

### 3.2 Effects of temperatures and irradiances

As the diatom community composition and photophysiological parameters were not significant between sites A, B and C, the diatoms collected from Site C (Exposed) were used for the incubation experiments. These communities were exposed to the highest irradiances and temperatures occurring naturally during the day. In general, the communities maintained at 30 ºC, 35 ºC, 40 ºC, 45 ºC and 50 ºC were still able to photosynthesize at all light levels. Statistical analysis indicated that the communities were mostly impacted by elevated temperature rather than irradiance. Control samples (30ºC and 35ºC) incubated at all irradiance levels had higher $F_v/F_m$ by comparison to other temperatures. There were no significant differences between either of these temperatures ($P < 0.05$, Table 5). The very low $F_v/F_m$ values indicate that severe photoinhibitive stress was present when the samples were exposed to high temperatures (55ºC and 60ºC) (Fig. 2a). The rETRmax, $\alpha$ and $E_k$ values also decreased rapidly as temperatures increased to 50°C (Fig. 2a-d) and no photosynthetic activity was observed at 55 ºC or 60 ºC at any irradiance level ($P < 0.05$, Table 5). Significant differences (ANOVA, $P < 0.05$) between temperature treatments were observed in all photosynthetic parameters. Although only limited photosynthetic activity was observed at higher temperatures, higher NPQ values were observed at these temperatures (Fig. 3), however, NPQ values were relatively low in all the treatments, especially at higher irradiances.

### 3.3 Recovery from temperature treatments

Elevated temperatures affected the recovery rates in the high temperature and irradiance treatments, ($P < 0.05$, Table 5). The highest recovery rate occurred at 40 °C. This corresponds with the maximum in-situ temperature at Site C, where the samples were collected from, was 42.5 ºC. The effects of the high temperatures were clearly visible at temperatures of 50 ºC, 55 ºC and 60 ºC, where severe depression of the recovery rate was observed (Fig. 4).

### 4 Discussion

The benthic diatom community in Tanjung Rhu estuary are naturally exposed to a high and variable light environment, and high temperatures caused by exposure at low tide. During low afternoon tide, the light intensity at the bottom (depth ~ 0.2 m) was measured at a maximum of 1900 µmol photons m$^{-2}$ s$^{-1}$ and decreased to a minimum of 170 µmol photons m$^{-2}$ s$^{-1}$ during high tide (depth ~ 1.0 m). Due to high water turbidity, these estuarine diatoms were only exposed to low irradiance during high tide. The surface water temperature increased from 28.5 ºC to 43 ºC during high to low tide. Even during the tidal cycle, the benthic diatom community could acclimatize to these changes by optimizing photosynthesis while minimizing photodamage caused by temperature and light fluctuations. Many benthic diatoms migrate into the sediment to avoid excess light (Serôdio et al., 2006a) as a photoprotective mechanism. Effects of migration were seen in significantly lower surface Chl $a$ biomass during low tide at all sampling sites. This ability to migrate is one reason why benthic diatoms are successful



in these environments, thus avoiding exposure to potentially damaging irradiance levels (Consalvey et al., 2004b). During high tide, turbidity decreases and the amount of light available for photosynthesis increases, resulting in diatoms staying on the sediment surface but migrating downward as light intensity becomes inhibiting. Also, during the rising tide (some samples were collected during flood tide) some diatoms were suspended into the overlying waters. In addition, higher temperatures, and the presence of macroinvertebrate grazers (personal observation) may have contributed to the lower

sediment surface chl $a$ biomass during low tide. Mitbavkar and Anil (2006) suggested that the nature of sediment is a significant factor determining both the abundance and composition of the diatom community. Higher biomass was generally recorded in muddy sediments than in sandier sediment (Ribeiro et al., 2013). This may explain why higher biomass was also found in the Tanjung Rhu system compared to a tropical shore in Penang, Malaysia. (McMinn et al., 2005)

Although rapid changes in temperature and irradiance occur naturally during tidal cycles in intertidal areas, the benthic diatom communities can acclimatize by minimizing photodamage through vertical migration (Serôdio et al., 2005) or physiological changes (e.g., diversion of excess energy away from photosystem reaction centers) (Consalvey et al. 2005, Coelho et al. 2011). Maximum quantum yield ($F_v/F_m$) values are often used as a sensitive indicator of photosynthetic stress (Du et al., 2018; McMinn et al., 2005). In a review by Campbell and Tyystjärvi (2012) it was suggested that for

photoinhibition measurements dark-adapted $F_v/F_m$ values can be used if measuring the rate of oxygen evolution is not possible. Thus, in this study, dark-adapted $F_v/F_m$ values were used as a proxy of diatom health/stress (Consalvey et al., 2005; Du et al., 2018) upon exposure to light and temperature stress. An increase in temperature and irradiance during low tide caused a decline in $F_v/F_m$ at all sites in the Tanjung Rhu estuary; these were low compared with optimum values of ~ 0.650 for healthy microalgae (McMinn and Hegseth, 2004). Similarly, low values were found by McMinn et al. (2005) on the

tropical shore of Muka Head and Songsong Island, Malaysia, where the average maximum quantum yields, which were only 0.325, were probably caused by either high irradiances or nutrient stress. Falkowski and LaRoche (1991) suggested that nutrient limitation would decrease the optimal quantum efficiency of phytoplankton. However, mangrove ecosystems are highly dynamic and support relatively high microalgae biomass and nutrient availability should not be a limiting factor (Hilaluddin et al., 2020; Rahaman et al., 2013). Nutrient concentrations (phosphate and nitrate) measured in this study were

within the average range for other tropical mangrove estuaries (Rajesh et al. 2001) and are unlikely to be limiting. Thus, it is likely that the high temperatures, of up to 42.5 °C during low tide in this study, were contributing to the low $F_v/F_m$ values.

Loik and Harte (1996) suggested that PSII reaction centers are susceptible to high temperatures and that the thermal stability of PSII can be influenced by the interaction between irradiance and high temperature since they will usually occur

simultaneously in the field. An increase in $F_v/F_m$ values may be attributed to a prolonged low light period during emersion at high tide. Prolonged exposure to high irradiances during low tide also affects $F_v/F_m$ values and reduces the algae's ability to fully recover during high tide. Du et al. (2018) also noted that NPQ and $F_v/F_m$ in benthic diatom were not impacted by low light <250 µmol photons m$^{-2}$ s$^{-1}$ by comparison to high light >500 µmol photons m$^{-2}$ s$^{-1}$. Furthermore, photoinhibition has





rarely been recorded in intertidal diatoms (Perkins et al. 2006). This is almost certainly due to their ability to migrate into the
sediment and away from damaging light (Consalvey et al. 2004a, Mouget et al. 2008). Their ability to subsequently recover
during high tide suggests that down-regulation of photosynthesis and up-regulation of photoprotection occurs, which
prevents serious damage to the photosystems during the high irradiances at low tide. Long and Humphries (1994) suggested
that photoinhibition that occurs during low tide, where $F_v/F_m$ declines, but recovers at high tide. However, this response was
not identified by Serôdio et al. (2008), who suggested that this short-term photoinhibition happened due to the incomplete
recovery of photosynthetic activity at low light and under nutrient depletion.

The reduction in in situ $F_v/F_m$ values at Tanjung Rhu were probably due to NPQ (non-photochemical quenching) mechanism
rather than photoinhibition that characterises the estuarine intertidal environment (Serôdio et al., 2008). During low tide, the
benthic diatom community reacted rapidly to the sudden increase in irradiance and temperature, protecting their photosystem
reaction centers by dissipating excess energy in the antenna complex of PSI and PSII (Goss and Lepetit, 2015). NPQ values
during low tides were higher than during high tides, suggesting that under high irradiance, the community compensated for
the possible over-excitation of the photosynthetic machinery through the dissipation of excess absorbed light through
xanthophyll-pigment-dependent energy dissipation. These communities were also able to modify their photophysiology by
maximizing their light-harvesting efficiency and hence had higher values of α at high tide than at low tide. $E_k$ values were
highest during low tide at Site C (Exposed), ($265.76 \pm 45.86$ μmol photons m$^{-2}$ s$^{-1}$), which indicates that the benthic diatom
was able to adapt to the changing light climate. Hence, at low tide the community modified its light harvesting capacity to
utilize the high irradiances to which they were exposed. Similarly, Perkins et al. (2006) found that benthic diatom grown in
high light had higher rETRmax and $E_k$ values, and lower α values. However, it should be noted that the length of light
exposure during RLCs affects these values, as short-term light exposure, e.g., 10 s in this study, is insufficient for complete
equilibration to each light level by comparison to P verse E curves, based on oxygen measurements (Torres et al., 2014).

### 4.1 Effects of irradiance and temperature incubations

Chlorophyll *a* fluorescence is an ideal tool for measuring the response of photosynthesis to changes in temperature, since
PSII is the most thermo-labile component of photosynthesis, and the majority of chlorophyll *a* fluorescence originates from
PSII under normal physiological conditions (Falkowski and Raven, 2007). Photosynthesis is extremely sensitive to stress
caused by intense temperatures and irradiance and it is often inhibited before other cellular functions are harmed.

When diatom dominated MPB samples were exposed to higher temperatures and irradiance, symptoms of irreversible
thermal damage were clearly visible at 55 °C and 60 °C. Limited photosynthetic activity at these temperatures suggests that
the cells would be suffering damaging effects from periodic increases in seawater temperatures greater than 50°C, especially
during low tide when the irradiance levels are also high. This suggests that high temperatures cause structural closure of the
PSII reaction centers and chloroplast dysfunction. Furthermore, at high temperatures and irradiances, light dependent



RUBISCO activation is inhibited, and this is closely correlated with a reversible reduction of $CO_2$ fixation. This is consistent with the decrease in the effective quantum yield, which displays a strong, quantitative relationship with the quantum yield of $CO_2$ assimilation (Oxborough and Baker, 1997).


Light and temperature exposure is expected to influence the sensitivity of response to changing light climate (Davison, 1991) during tidal cycles and hence affect values of photosynthetic efficiency (α). However, α can also be impacted by temperature changes, which can have a strong influence on the photosynthetic capacity (Claquin et al., 2008). The α values observed here showed a significant decline when exposed to high irradiances (up to 1800 μmol photons m$^{-2}$ s$^{-1}$) and

temperatures (up to 60 ℃). The reduction in α as temperature increased is consistent with the results reviewed by Davison (1991), where α also declined as temperature increased. The effects of temperatures on α values were only significant when the cells were exposed to both adverse irradiances and temperatures. Defew et al. (2004) showed that in temperate microalgae, there were no clear trends in α values, with either temperature or light, when samples were exposed to temperatures ranging from 10 ℃ to 26 ℃ and irradiance up to 350 μmol photons m$^{-2}$ s$^{-1}$. This variation in α response to

temperature between studies is probably a result of thermal acclimation being species dependent (Claquin et al. 2008) and also that major impacts only occur at the extremes. Fanesi et al. (2016) also discussed how temperature affected absorbed light in different freshwater phytoplankton groups but did not examine the effects at temperature extremes. Furthermore, some of the variation could also have been caused by the fluorescence method itself and the duration of RLCs exposure. For example, Lefebvre et al. (2011) noted that α was systematically lower due to the slower relaxation of NPQ in 10 s RLCs.


The $E_k$ index provides an indication of the irradiance at which energy is diverted from photochemistry to heat dissipation. It represents the degree of acclimation to the ambient light climate (Schreiber, 2004). $E_k$ is used as an indicator of the photoacclimation status of photosynthetic organisms. Higher values of $E_k$ suggest acclimation to higher irradiances (McMinn et al. 2005). Benthic diatoms exposed to high irradiance herein (1800 μmol photons ·m$^{-2}$ ·s$^{-1}$) were not able to adjust their

metabolism to maximize their response to these irradiances. Similarly, in their natural habitat, during low tide the *in-situ* data showed that the samples were light saturated at an *in-situ* irradiance of 1900 μmol photons m$^{-2}$ s$^{-1}$. However, the $E_k$ values recorded at all experimental irradiances and temperatures (40 ℃ to 50 ℃) were similar to the *in-situ* $E_k$ values. When higher temperatures (above 50 ℃) were imposed, significant declines in $E_k$ were observed. Falkowski and Raven (2013) suggested that as temperatures and irradiances increase, $E_k$ also increases. This suggests that these temperatures (above 50℃) were

approaching a critical temperature threshold and affecting the cells' capability to photosynthesize. These effects were also observed in the rETRmax values.

The rETRmax parameter has been used in many recent benthic diatom studies (e.g Cartaxana et al. 2011, Du et al. 2018) as it is closely related to the maximum photosynthetic capacity, which is obtained when the rate of photosynthesis is limited by





the activity of the electron transport chain or Calvin cycle enzymes (Ralph and Gademann 2005). The rETRmax values recovered here were typical of diatom responses to high irradiances, where, as irradiance increases, photosynthetic rates become increasingly nonlinear and rise to a saturation level (Blanchard et al. 2004). Perkins et al. (2006) suggested that rETRmax determined after 10s of actinic light during RLCs are lower when cells have been previously acclimated to higher irradiance. However, to minimize the potential for the RLCs irradiance to cause photoacclimation the RLCs duration should

be as short as possible (Ralph and Gademann, 2005). Unlike *in-situ* studies, where natural biochemical and physical gradient are preserved, experiments such as those described here, where cells are suspended, will give a slightly different result. Higher $E_k$ and rETRmax are typical of high irradiance-acclimated samples, where the cells have modified their light harvesting pigment to utilize the irradiance to which they were exposed. Benthic microalgae acclimated to low irradiance are able to modify their physiology and maximize their light harvesting efficiency, thus providing higher values of α. These

characteristics were observed in both the low and high irradiance only at ambient temperatures (up to 45 ºC). The photosynthetic rate declined when samples were exposed to elevated temperatures. Severe effects were seen in higher irradiance experiments. Here, as the temperatures increased above 55 ºC, these algae appeared to be affected by thermal stress and were not able to acclimate to the high temperatures.

Although being exposed to variations in irradiance and temperature during tidal cycles, benthic diatoms have a range of mechanisms to facilitate and optimize light interception and utilization. Here, cells exposed to high irradiances and temperatures of 40ºC did not show any sign of photoinhibition as they were naturally growing at temperatures up to 43 ºC during low tide exposure. The cells were able to dissipate the excess light via NPQ or possibly migration. Although migration was not investigated in this study, behavioural photoprotection by vertical migration is a common response to

avoid high light by pennate diatoms (Cartaxana et al., 2011; Du et al., 2018; Mitbavkar and Anil, 2002). In contrast, samples exposed to the same irradiance level, but higher temperatures (55 ºC and 60 ºC) were not able to protect themselves from photodamage and photoinhibition. Exposure to high irradiances coupled with high temperatures caused an excess of absorbed light energy that was unable to be used in the photosynthetic process, resulting in the increases in NPQ values observed in this study. Although no photosynthetic activity was present at the highest temperatures (55 ºC and 60 ºC), heat

dissipation was still active. Xanthophyll cycling is especially significant when short term changes in the light climate occurs (seconds to minutes), such as those happening in the Tanjung Rhu intertidal sediments. Samples incubated in the dark and at high temperatures (55 ºC and 60 ºC) were seen to have higher NPQ levels. In addition, these samples showed sudden increments in NPQ from 50 ºC to 55 ºC. However, as no photosynthetic activity was observed at 55 ºC, it is possible that the NPQ signal was generated from dead diatom cells. Nonetheless, the ability of dark-adapted diatoms to develop NPQ has

been considered an adaptive advantage, providing a method to prevent degradation of light harvesting pigments (Serôdio et al. 2006b, Lavaud and Lepetit, 2013). Hence further investigation is required to better understand the impact of extreme temperatures on the mechanism of NPQ, formation of reactive oxygen species, photodamage, and ultimately inactivation of PSII.



The ability of plants to recover from stress is a major characteristic determining their ability to survive under hostile conditions (Wu et al., 2017, Ralph and Gademann, 2005). In this study, at higher temperatures, there are indications that degradation of D1 proteins may have occurred and that it disrupted the recovery of PSII, thus preventing cells from adapting to the ambient conditions. At 40 ºC, dark acclimated samples were able to recover faster than at higher temperatures, suggesting that high irradiances (1800 and 890 µmol photons $\cdot m^{-2} \cdot s^{-1}$) increases PSII damage and contributes to the slower

recoveries. In addition, these sample had been exposed to similar temperatures to those *in-situ* samples during low tide, thus, suggesting that combined high light and temperature could have a more significant than of temperature alone. These experiments showed that samples that were exposed to lower temperatures, had a better recovery rate than those exposed to higher temperatures (>55 ºC). The lack of recovery at temperatures higher than 55 ºC suggests that the photosynthetic enzymes had been inactivated or damaged, and acclimation to these temperatures was not possible. Inhibition of

photosynthesis is reversible when temperatures are only a little higher than optimal (moderate heat stress), whereas damage to the photosynthetic apparatus is permanent under severe heat stress (Berry and Bjorkman 1980). At low tide in tropical estuaries water temperatures regularly exceed 50 °C, temperatures usually considered outside the range of survival of marine autotrophs. With predicted global warming these conditions are likely to become both more frequent and more extreme (Stocker et al. 2014). The 50 °C environmental water temperature seems to be a tipping point, above which photosynthesis is

irreparably impaired. In these circumstances, it is likely that tropical estuarine benthic microalgal community composition will change, and benthic primary production will decrease. This will likely have a cascading effect throughout these ecosystems.

**Conclusions**

   While the data presented herein demonstrates the response of tropical benthic diatoms to high irradiance and extreme

temperature, the use of RLC data alone prevents a more rigorous examination of non-photosynthetic quenching activities such as the xanthophyll cycle (Cartaxana et al., 2013). Nonetheless, data presented here provides an insight into the capacity of tropical diatom dominated MPB communities to tolerate these environmental stresses, and the degree to which those stresses could damage the photosynthetic apparatus. Thus, parameters derived from RLCs were useful to indicate the photosynthetic capacities over a series of temperature and light, even though they are derived from measurements over a

range of irradiances. Likewise, since most previous studies of the combined impacts of high temperature on benthic communities have focussed on light tropical and temperate communities, responses of tropical communities are unknown and is likely to vary significantly. Short-term changes (hours) in temperatures up to 45 ºC, such as experienced by the community in Tanjung Rhu estuary, will not severely affect their photosynthetic apparatus, but higher temperatures (55 ºC and 60 ºC) will cause severe damage and irreparable photoinhibition. The responses of benthic microalgae to light are also

dependent on the temperature regime under which they are exposed. The benthic diatoms in Tanjung Rhu experience these



effects during tidal emersion and during the night period. In summary, optimal temperatures for benthic microalgae photosynthetic performance is up to 45 ºC, beyond this temperature the PSII function will be severely damaged and could lead to chronic photoinhibition.

## Author contribution

**Sazlina Salleh:** Conceptualization, Validation, Formal anlysis, Investigation, Data Curation and Writing-Original Draft. **Andrew McMinn**: Validation, Resources, Writing – Review & Editing, Visualization and Supervision.

## Competing interests

The authors declare that they have no conflict of interest.

## Acknowledgements

We would like to thank University of Tasmania and Universiti Sains Malaysia for the logistic support and funding.

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





**Tables and Figures:**


**Table 1: Site description of *in-situ* environmental and fluorescence parameters data collection.**

| Site | Description |
|---|---|
| Site A: Submerged | This site is located at the middle to the Tanjung Rhu estuary. The sediment collected from this area are constantly being submerged during low tide with an average water level of ~ 0.2 meters. |
| Site B: Shaded | This site is located at the estuary bank (inner section) and is shaded by mangroves trees. During low tide the sediment is exposed and partially dried without any seawater. |
| Site C: Exposed | This site is located at the estuary bank (outer section) without any trees or any shaded structure. At low tide the sediment is exposed and partially dried without any seawater. |

**Table 2: Average values (n=3) of the pore-water physical variables measured during high and low tide at Site A, B**
**and C. Data are means ± SD, *n = 3*.**

| Parameters | Site A: Submerged | | Site B: Shaded | | Site C: Exposed | |
|---|---|---|---|---|---|---|
| | Low tide | High tide | Low tide | High tide | Low tide | High tide |
| Sampling time (UTC +8) | 1500 - 1700 | 1000 - 1200 | 1500 - 1700 | 1000 - 1200 | 1500 - 1700 | 1000 - 1200 |
| Light level ($\mu mol$ photons $\cdot m^{-2} \cdot s^{-1}$) | 932.38 ± 57.11 | 244.60 ± 6.37 | 662.77 ± 101.84 | 173.42 ± 13.73 | 1978.27 ± 99.68 | 273.60 ± 24.37 |
| Temperature (ºC) | 40.1 ± 0.11 | 29.2± 0.12 | 35.0 ± 0.06 | 28.5 ± 0.06 | 42.5± 0.05 | 29.3 ± 0.05 |
| Salinity | 33.0 ± 0.0 | 33.0 ± 0.0 | 33.0 ± 0.0 | 33.0 ± 0.0 | 34.0 ± 0.0 | 33.0 ± 0.0 |
| Nitrate (mg/L) | 0.325 ± 0.020 | 0.431 ± 0.047 | 0.319 ± 0.045 | 0.326 ± 0.034 | 0.299 ± 0.046 | 0.346 ± 0.059 |
| Phosphate (mg/L) | 0.206 ± 0.011 | 0.462 ± 0.057 | 0.216 ± 0.022 | 0.405 ± 0.049 | 0.139 ± 0.044 | 0.377 ± 0.064 |
| Ammonium (mg/L) | 0.001 ± 0.001 | 0.005 ± 0.004 | 0.001 ± 0.000 | 0.002 ± 0.001 | 0.002 ± 0.001 | 0.001 ± 0.000 |





**Table 3: In-situ photosynthetic parameters of benthic diatom at Tanjung Rhu estuary (Site A, B and C) in the surface 10mm (depth ~ 0.2 m at low tide and 1.0 m at high tide). Data are means ± SD, *n = 3.***

| Parameters | Site A: Submerged | | Site B: Shaded | | Site C: Exposed | |
|---|---|---|---|---|---|---|
| | Low tide | High tide | Low tide | High tide | Low tide | High tide |
| Chlorophyll *a* (mg chl *a* m$^{-2}$) | 17.64 ± 1.34 | 35.08 ± 2.95 | 21.22± 6.78 | 37.38 ± 8.16 | 10.23 ± 1.73 | 20.19 ± 3.06 |
| $F_v/F_m$ (maximum quantum yield) | 0.195 ± 0.062 | 0.346 ± 0.099 | 0.216 ± 0.061 | 0.334 ± 0.089 | 0.192 ± 0.018 | 0.318 ± 0.148 |
| $\Delta F/F_{m'}$ (effective quantum yield) | 0.170 ± 0.040 | 0.317 ± 0.133 | 0.148 ± 0.038 | 0.223 ± 0.021 | 0.182 ± 0.051 | 0.273 ± 0.077 |
| rETRmax | 16.017 ± 8.222 | 21.038 ± 3.215 | 15.723 ± 5.403 | 20.921 ± 1.799 | 20.857 ± 3.029 | 23.815 ± 4.391 |
| $E_k$ (µmol photons · m$^{-2}$ · s$^{-1}$) | 187.44 ± 33.17 | 173.17 ± 38.81 | 240.70 ± 85.92 | 185.95 ± 52.02 | 265.76 ± 45.86 | 192.77± 39.19 |
| $\alpha$ | 0.082 ± 0.028 | 0.128 ± 0.042 | 0.066 ± 0.011 | 0.119 ± 0.037 | 0.079 ± 0.005 | 0.129 ± 0.043 |
| NPQ | 0.561 ± 0.237 | 2.336 ± 0.755 | 1.608 ± 0.545 | 1.375 ± 0.625 | 0.315 ± 0.050 | 1.861 ± 0.987 |





**Table 4.  Two-way ANOVA for (a) physical parameters (bottom irradiance (PAR), temperature, nitrate, phosphate, ammonium and Chl a; b. Photosynthetic parameters (effective quantum yield, $\Delta F/F_m$'; maximum quantum yield, $F_v/F_m$; relative maximum photosynthetic rate, rETRmax; light saturation parameter, $E_k$; photosynthetic efficiency, $\alpha$ and, non-photochemical quenching, NPQ. with respect to tide (low and high) and site (A, B and C) .**

| Factor | (a) Environmental Parameters | | | | | |
|---|---|---|---|---|---|---|
| | **PAR** | **Temperature** | **Nitrate** | **Phosphate** | **Ammonium** | **Chl a** |
| Tide | **<0.001** | **<0.001** | **0.023** | **<0.001** | 0.179 | **0.000** |
| Site | **<0.001** | **<0.001** | 0.073 | **0.035** | 0.227 | **0.001** |
| Tide x Site | **<0.001** | **<0.001** | 0.189 | 0.426 | 0.062 | 0.376 |

| Factor | (b) Photosynthetic Parameters | | | | | |
|---|---|---|---|---|---|---|
| | **$F_v/F_m$** | **$\Delta F/F_m$'** | **rETRmax** | **$E_k$** | **$\alpha$** | **NPQ** |
| Tide | **0.007** | **0.009** | 0.077 | **<0.001** | **0.006** | **0.004** |
| Site | 0.718 | 0.990 | 0.302 | 0.035 | 0.754 | 0.483 |
| Tide x Site | 0.820 | 0.880 | 0.905 | 0.426 | 0.975 | **0.030** |

Significant difference ($P<0.05$) indicated in bold.





**Table 5 : Two way ANOVA of effects of temperature (0 ºC to 60 ºC) and irradiance level (1800, 890 and 0 µmol photons m⁻² s⁻¹) on the photosynthetic parameters (F$_v$/F$_m$, maximum quantum yield; α, photosynthetic efficiency; rETRmax, relative electron transport rate; E$_k$, photoacclimation index), NPQ (non-photochemical quenching) and recovery rate the benthic diatom after exposure to temperatures treatment for 1 hour.**


| Parameter | Factor | df | F | Sig. |
|---|---|---|---|---|
| F$_v$/F$_m$ | Temperature | 6 | 50.350 | **<0.001** |
| | Irradiance | 2 | 0.124 | 0.884 |
| | Temperature x Irradiance | 12 | 0.611 | 0.821 |
| rETRmax | Temperature | 6 | 145.839 | **<0.001** |
| | Irradiance | 2 | 28.269 | **<0.001** |
| | Temperature x Irradiance | 12 | 7.400 | **<0.001** |
| α | Temperature | 6 | 48.487 | **<0.001** |
| | Irradiance | 2 | 0.693 | 0.506 |
| | Temperature x Irradiance | 12 | 0.691 | 0.751 |
| E$_k$ | Temperature | 6 | 88.619 | **<0.001** |
| | Irradiance | 2 | 19.144 | **<0.001** |
| | Temperature x Irradiance | 12 | 5.656 | **<0.001** |
| NPQ | Temperature | 6 | 4.813 | **<0.001** |
| | Irradiance | 2 | 0.369 | 0.694 |
| | Temperature x Irradiance | 12 | 1.382 | 0.212 |
| Recovery Rate | Temperature | 6 | 37.111 | **<0.001** |
| | Irradiance | 2 | 1.836 | 0.172 |
| | Temperature x Irradiance | 12 | 8.107 | **<0.001** |

Significant difference ($P<0.05$) indicated in bold.


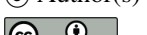



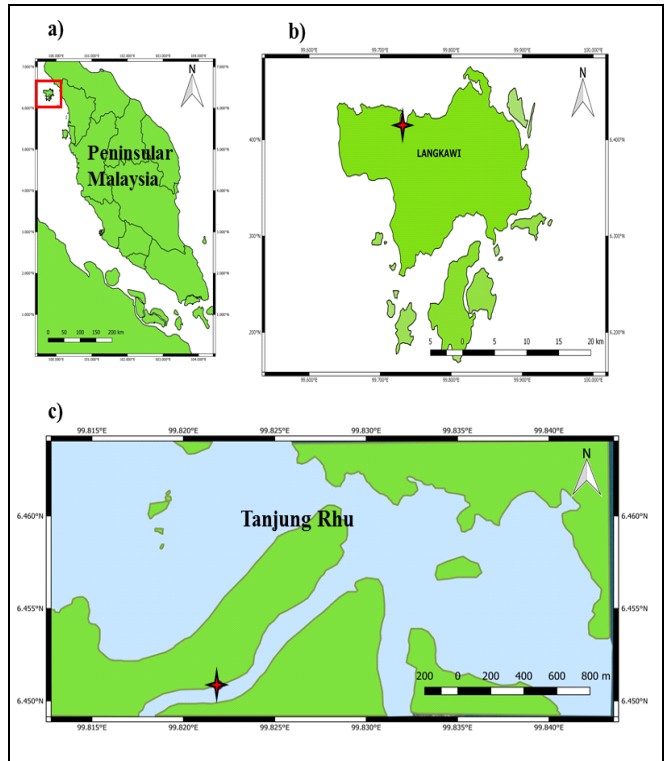

**Fig. 1: Location of the sampling station in Tanjung Rhu estuary, Langkawi, Malaysia. a) Map of Peninsular Malaysia. The square indicates Pulau Langkawi, b) Map of Pulau Langkawi and c) Map of Tanjung Rhu. The red star indicates the sampling area.**






**Fig. 2: Derived photosynthetic parameters of RLC.** Parameters were plotted against experimental temperatures (30, 35, 40, 45, 50, 55 and 60 °C) at irradiances of 1800 µmol photons m$^{-2}$ s$^{-1}$ (close square), 890 µmol photons m$^{-2}$ s$^{-1}$ (close circle) and 0 µmol photons m$^{-2}$ s$^{-1}$ (open triangles). a) Maximum quantum yield, b) Photosynthetic efficiency (α), c) rETRmax and d) Photoacclimation index, (Ek). Values are means ± SD ($n = 3$).





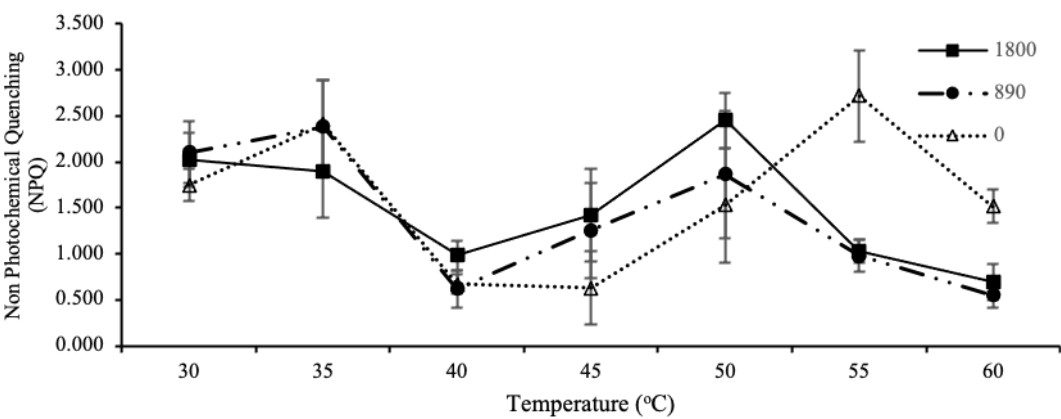


**Fig. 3: The NPQ of samples at each irradiance level (1800 photons m$^{-2}$ s$^{-1}$ (close square), 890 photons m$^{-2}$ s$^{-1}$ (close circle) and 0 photons m$^{-2}$ s$^{-1}$ (close triangles)) and temperatures (30, 35, 40, 45, 50, 55 and 60 ºC). NPQ was obtained at the end of the RLCs with actinic irradiance of 1554 µmol photons m$^{-2}$ s$^{-1}$. Values are means ± SD ($n = 3$).**

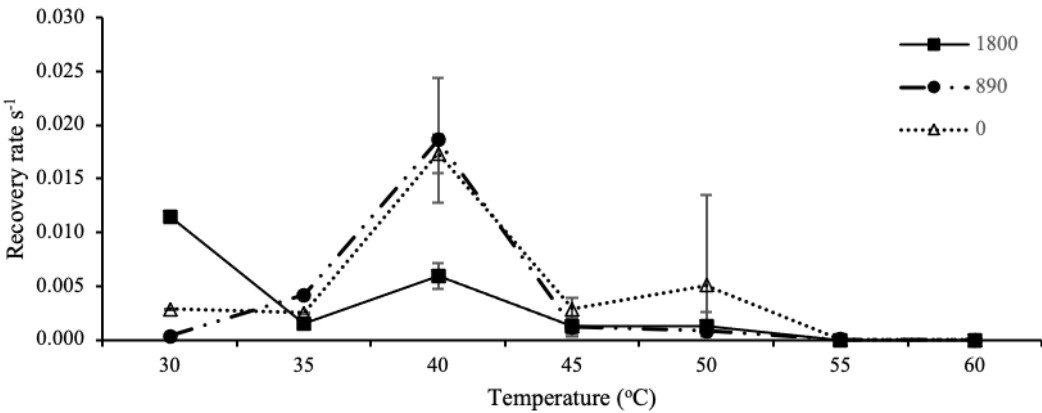


**Fig. 4: Recovery rate s$^{-1}$ of F$_v$/F$_m$ after the RLC for low irradiance experiments. Recovery rates were plotted against experimental temperatures at irradiances of (1800 µmol photons m$^{-2}$ s$^{-1}$ (close square), 890 µmol photons m$^{-2}$ s$^{-1}$ (close circle) and 0 photons m$^{-2}$ s$^{-1}$ (open triangles)) and temperatures (30, 35, 40, 45, 50, 55 and 60 ºC). In this recovery analysis, samples of 30 ºC and 35 ºC were analyzed without replicates; thus, no error bar was obtained.**
