# Peer review of "Tolerance of tropical marine microphytobenthos to elevated irradiance and temperature"

_Biogeosciences, 2021_

## Referee Comment (RC1)

**Review Report:**

Response of tropical marine benthic diatoms exposed to elevated irradiance and tempperature

**General comment:**

The manuscript proposal and experimental design are good. This is an interesting study to evaluate the response of tropical benthic diatoms to high irradiance and temperature based on chl a and photosynthetic parameter analysis. Therefore, this manuscript is recommended for publication. However, there are minor correction need to be done. Please refer to the comments given below:

**Abstract**
Abstract was good, but what is the unit for recovery rates?
"The recovery rates for samples exposed to temperatures of 40$^{\circ}$C  were 4.01E-03 ± 0.002 (kindly put the unit here, If any) and decreased to 1.01E-05 ± 0.001 (kindly put the unit here, if any) at 60$^{\circ}$C

**Introduction:**
Introduction is good

**Materials and methods**
Materials and methods are  good

**Results**
Results are  good

**Discussion**
Discussions are good

**Conclusions**
Conclusions are good.

**References**
References are good.
However if possible kindly update the citation/references and it should be the most current within 5 years

**Figure**
For Figure 2 (a, b, d), kindly put the x-axis title or you can put the x-axis title at the last graph (Fig.2d) only

---

## Referee Comment (RC2)

**Aninda Mazumdar**
**Associate Editor to Biogeosciences**
Title: Response of tropical marine benthic diatoms exposed to elevated irradiance and temperature
Author(s): Sazlina Salleh and Andrew McMinn
MS No.: bg-2021-18
MS type: Research article
Iteration: Initial Submission

**DEAR ANINDA, PLEASE FIND IN THE FOLLOWING MY CRITIC AND RECOMENDATION ON THE ASIGNED PAPER.**
**MY OPINION IS THAT: THIS MANUSCRIPT HAS GOOD POTENTIAL IN TERMS OF ECOPHYSIOLOGICAL ECOLOGY AND SHOULD BE EVENTUALLY PUBLISHED. HOWEVER, IT HAS TO BE MODIFIED ACCORDING TO THE OBSERVATIONS I HAVE LISTED BELOW.**

**TITLE: Response of tropical marine benthic diatoms exposed to elevated irradiance and temperature**

**SEEMS IMPRECISE...TRY:**
TOLERANCE/STRATEGIES OF TROPICAL BENTHIC DIATOM ASSEMBLAGES TO HIGH IRRADIANCE AND TEMPERATURE
**IT CORRESPONDS MUCH BETTER WITH THE DERIVED CONCLUSION**

**Abstract.**
**LINE 12** Shallow tropical marine environments are likely to experience future water temperatures that will challenge the ability of life to survive.
**FUTURE RISE IN WATER TEMPERATURE? HOW FAR INTO THE FUTURE….15 YEARS (NOW)?**
**THIS PREMISE LACKS SCIENTIFIC OR PHILOSOPHICAL (LOGIC) BASIS**

Changes in temperature and irradiance during tidal cycles in the Tanjung Rhu estuary, Langkawi,Malaysia in 2007 did not significantly affect the benthic diatom communities
**NOT COMMUNITIES BUT TAXOCENOSES. WHAT TYPE OF CHANGES WERE EXPECTED?**

**LINE 22** These characteristics suggest that the benthic diatom communities in this estuary are already suffering from thermal damage
**IN GENERAL IT CONTRADICTS ADAPTATION PRINCIPLES THAT DRIVE EVOLUTIONARY ECOLOGY, INASMUCH THE SURVEYED DIATOM ASSOCIATIONS ARE ADAPTED. THUS THE FOLLOWING SENTENCE LOOSES MEANING**
and that enhanced photoinhibition would result if exposed to elevated temperatures, especially during low tide. 50 °C is apparently a temperature threshold for tropical 25 estuarine **??** benthic microalgal communities.
**NOT A CLEAR SENTENCE...**
Future warming is likely to cause this temperature to occur more frequently, which will cause a reduction in benthic primary production
**TO MUCH CONFIDENCE ON THE GLOBAL HYPERWARMING EXPECTATIONS MAKE THIS CONCLUDING REMARK OUT OF PLACE (DISCRETE DATA CONTRADICTING ECOLOGICAL PLANETARY EVOLUTION)**

**INTRODUCTION**
LINE 65 2005; Underwood, 2002). However, there are relatively few studies of benthic diatoms in tropical estuaries (Patil and Anil 2008). Although the photosynthesis of tropical benthic diatoms is likely to be highly sensitive to elevated temperature,comparatively little attention has been given to the combined effects of both elevated light and temperature. Water temperatures in tropical

marine ecosystems are already high and relatively small increases can have severe negative impacts. Intertidal and subtidal communities are particularly vulnerable and increasing temperatures, resulting from global warming, will have unpredictable consequences.
LINE 70 This study examines the impact of high light and temperature as temperatures increase. These conditions are predicted to worsen with the increase in seawater temperatures resulting from climate change.

**A STUDY SUCH AS THIS CAN NOT DO WITHOUT A HYPOTHESIS. I SUGGEST TO MAKE IT EXPLICIT ON THE BASIS OF, HOW DO BENTHIC DIATOMS FROM TROPICAL SHOW ADAPTATIONS TO HIGH TEMPERATURE AND IRRADIANCE. PLEASE FORGETT ABOUT GLOBAL WARMING, WHICH BESIDES BEING A SWINDLE (HARSH DISCUSSION) IT REFERS TO PLANETARY ECOLOGY AND THE RELATION WITH YOUR STUDY IS VERY REMOTE-**
**JUST KEEP IT ECOPHYSIOLOGICAL, IT HAS A VLUE OF ITS OWN**

**MATERIAL AND METHODS**
**LINE 80** mangrove forest, mainly dominated by Rhizophora and Avicennia species. The benthic habitat is mostly composed of coarse sand with patches of mud. The water normally carries a high-suspended load, originating partly from the mangrove forest and from the river sediment itself. To gain a better understanding of the surrounding environment, samples were collected in April 2007 from three sites (Sites A, B and C) all in proximity (20 meters) to the intertidal zone of the Tanjung Rhu Estuary, during both high and low tide.
**HIGHLIGHTED SENTENCE IS VAGUE. SHOULD BE ERASED OR EXTENDED WITH PRECISE DESCRIPTION OF HOW A SUCH A BETTER UNDERSTANDING IS PURSUED.**
**ALSO, SAMPLES FROM 2007? AN EXPLANATION FOR THIS ASYNCHRONY IS REQUIRED**

**LINE 85** Description of sites A, B and C are provided in Table 1. Environmental data (salinity, temperature) and photosynthetic parameters were collected at ebb and flood tide. Water height at ebb tide was approximately 0 to 0.2 m and 0.5 to 1 m at flood tide. On each occasion, seven 15 mm diameter hand-pushed sediment cores were taken; three for photosynthetic parameter analysis, three for chlorophyll a analysis and one for species composition. For the temperature incubation experiments, the top 10 mm (approximately) of the sediment was manually scraped off

**LINE 90** and placed in a dark plastic bag (20 x 90 20 cm). Samples were stored in the dark and promptly returned (approximately 15 min) to the laboratory for the experiments.
At each sampling site, water quality measurements (temperature, salinity, and photosynthetically active irradiance (PAR) were measured using a Hydrolab Datasonde 4a (Hach, Loveland CO). Water samples for all parameters and nutrient analyses were collected during high tide (water level: 1.0 m) and low tide (water level: 0.2 m).
**IT APPEARS REPEATED, HOWEVER YOU ARE REFERING TO HYDROLOGICAL VARIABLES, NOT TO WATER QUALITY WHICH INDICATES POLLUTION DEGREE**

**METHODS**
**LINE 199 STATE IF DATA WERE TESTED FOR HOMOSCEDASTICITY AND NORMAL DISTRIBUTION BEFORE APPLYING PARAMETRIC STATISTICAL TESTS, AND IF SO, WHAT TEST WAS USED**

**RESULTS**
**ACCORDING TO WHAT IT IS READ IN (METHOD) ON COMMUNITY COMPOSITION (LINES 100-110), WHAT IT IS PRESENTED AS RESULT:**
**(LINE 195)** The diatom genera Cocconeis, Navicula and Rhopalodia dominated the benthic habitats, which predominantly consisted of course sand with mud/silt patches. The sediment characteristics were similar among the sampling sites. The most common species found were Cocconeis placentula, Navicula raphoneis, Navicula clipeiformis and Rhapalodia acuminatea. The composition of the diatom communities

at all sites and tides were similar without any differences in dominant species.
**IS NOT BASED ON PRESENTED EVIDENCE, SUCH AS SPECIES LIST, IMAGES, RELATIVE ABUNDANCES SIMILARITY MEASUREMENTS**
**HENCE, IT WOULD BE ADEQUATE TO REMOVE ANY  PARTICULARITY ON COMMUNITY ANALYSIS, BOTH IN METHOD AND RESULTS**
**I SUGGEST TO FURTHER WORK SAID DATA ON DIATOMS AN TRY TO PUBLISH INDEPENDENTLY**

**LINE 206** The highest nutrient levels were recorded during high tide, when seawater re-entered the estuary (Table 2 and Table 4(a)) There was a least a 50% increase in nitrate and phosphate levels during high tide, although the changes were not significant (Table 2).
**MAYBE BECAUSE THE STATISTICAL TESTS USED (PARAMETRIC) WERE NOT APPROPIATE?**

**DISCUSSION**
**LINES 240-258 DO NOT PRESENT ANY INFERENTIAL CONTRIBUTION (THEORY); MOST LIKELY BECAUSE NO WORK WAS DONE ON A MOST NEEDED HYPOTHESIS**

**LIKEWISE**
**LINES 260-285 WOULD DO BETTER CONTRASTING A HYPOTHESIS AND NOT JUST COMPARING WITH OTHER STUDIES, WHICH HAS TO BE DONE WHEN STATING THE PROBLEM/HYPOTHESIS**

**LINES 286-389 I FEEL ARE ON THE RIGHT TRACK IN TERMS OF ECOPHYSIOLOGICAL STRATEGIES THAT EXPLAIN THE PERMANENCE AND ABIDENCE OF DIATOM POPULATIONS UNDER THESE CONDITIONS**

**LINE 390 DIATOMS ARE NOT PLANTS!**

**CONCLUSION**
**I FIND CONCLUSION WELL DERIVED, BUT IN AGREEMENT WITH THE ECOPHYSIOLOGICAL STRATEGY VIEW OF DIATOMS, AND NOT IN THE TERMS THAT I HAVE OBSERVED WHICH SHOULD BE AVOIDED GLOBAL WARMING OR BYPASSING THE FACT THAT DIATOM POPULATIONS ARE WELL ADAPTED TO EXTREME CONDITIONS, WHETHER TROPICAL AS THESE OR IN DESERT SABKAHS.**
**HOWEVER, THE STARTING SENTENCE SHOULD BE ELIMINATED:**

**LINE 405 While the data presented herein demonstrates the response of tropical benthic diatoms to high irradiance and extreme temperature, the use of RLC data alone prevents a more rigorous examination of non-photosynthetic quenching activities such as the xanthophyll cycle (Cartaxana et al., 2013). Nonetheless,**

---

## Author Comment (AC2)

**RESPONSES TO REVIEWERS' COMMENTS**
Manuscript ID: bg-2021-18

*Reviewer 2*

**TITLE: Response of tropical marine benthic diatoms exposed to elevated irradiance and temperature**
**SEEMS IMPRECISE...TRY:**

**TOLERANCE/STRATEGIES OF TROPICAL BENTHIC DIATOM ASSEMBLAGES TO HIGH IRRADIANCE AND TEMPERATURE**
**IT CORRESPONDS MUCH BETTER WITH THE DERIVED CONCLUSION**

Response:

Thank you for the suggestion. Title changed as suggested. *New Title: Tolerance of tropical marine microphytobenthos to elevated irradiance and temperature*

**Abstract.**
**LINE 12 Shallow tropical marine environments are likely to experience future water temperatures that will challenge the ability of life to survive.**
**FUTURE RISE IN WATER TEMPERATURE? HOW FAR INTO THE FUTURE....15 YEARS (NOW)?**
**THIS PREMISE LACKS SCIENTIFIC OR PHILOSOPHICAL (LOGIC) BASIS**

Response:

We disagree that references to climate change should be removed from this manuscript. Climate change provides the underlying reason for undertaking this research. Examining the effects of elevated temperatures, such as those predicted to occur this century, is a completely reasonable and legitimate approach.

**Changes in temperature and irradiance during tidal cycles in the Tanjung Rhu estuary, Langkawi, Malaysia in 2007 did not significantly affect the benthic diatom communities**
**NOT COMMUNITIES BUT TAXOCENOSES. WHAT TYPE OF CHANGES WERE EXPECTED?**

Response:

During the field monitoring, we expected to see change in the diatom assemblage composition, however this was not observed. We have mostly replaced 'benthic diatom communities' with microphytobenthos (MPB), as the sediment samples would have contained many species other than diatoms. Please see Abstract Line 13.

Line 13:"The photosynthetic productivity of tropical microphytobenthos (MPB) is largely driven by changes in light intensities and temperature at the surface of sediment flats during emersion. Changes in temperature and irradiance during tidal cycles in the Tanjung Rhu estuary, Langkawi, Malaysia in 2007 significantly affected the photosynthetic capacities of the MPB."

**LINE 22 These characteristics suggest that the benthic diatom communities in this estuary are already suffering from thermal damage**
IN GENERAL IT CONTRADICTS ADAPTATION PRINCIPLES THAT DRIVE EVOLUTIONARY ECOLOGY, INASMUCH THE SURVEYED DIATOM ASSOCIATIONS ARE ADAPTED. THUS THE FOLLOWING SENTENCE LOOSES MEANING **and that enhanced photoinhibition would result if exposed to elevated temperatures, especially during low tide. 50 °C is apparently a temperature threshold for tropical estuarine** ?? **benthic microalgal communities.**
NOT A CLEAR SENTENCE...

Response:

Sentence has been clarified. See Line 25 onwards. 50°C refers to temperature threshold for photosynthetic performance of benthic diatom suggested from the outcome of this study.

Line 25: These characteristics suggest that the MPB communities in this estuary were able to adapt to temperature variation. However, enhanced photoinhibition would result if exposed to elevated temperatures, especially during low tide where in situ temperature was already 43°C. Hence, if in situ temperature was to further increase during tidal emersion, 50 °C could be a temperature threshold for photosynthetic performance of tropical estuarine benthic microalgal communities.

**Future warming is likely to cause this temperature to occur more frequently, which will cause a reduction in benthic primary production**
TO MUCH CONFIDENCE ON THE GLOBAL HYPERWARMING EXPECTATIONS MAKE THIS CONCLUDING REMARK OUT OF PLACE (DISCRETE DATA CONTRADICTING ECOLOGICAL PLANETARY EVOLUTION)

Response:

We disagree with the reviewer about the consequences of global warming and intend to leave these sentences in.

**INTRODUCTION**
**LINE 65 2005; Underwood, 2002). However, there are relatively few studies of benthic diatoms in tropical estuaries (Patil and Anil 2008). Although the photosynthesis of tropical benthic diatoms is likely to be highly sensitive to elevated temperature, comparatively little attention has been given to the combined effects of both elevated light and temperature. Water temperatures in tropical marine ecosystems are already high and relatively small increases can have severe negative impacts. Intertidal and subtidal communities are particularly vulnerable and increasing temperatures, resulting from global warming, will have unpredictable consequences.**

**LINE 70 This study examines the impact of high light and temperature as temperatures increase. These conditions are predicted to worsen with the increase in seawater temperatures resulting from climate change.**

A STUDY SUCH AS THIS CAN NOT DO WITHOUT A HYPOTHESIS. I SUGGEST TO MAKE IT EXPLICIT ON THE BASIS OF, HOW DO BENTHIC DIATOMS FROM TROPICAL SHOW ADAPTATIONS TO HIGH TEMPERATURE AND

**IRRADIANCE. PLEASE FORGETT ABOUT GLOBAL WARMING, WHICH BESIDES BEING A SWINDLE (HARSH DISCUSSION) IT REFERS TO PLANETARY ECOLOGY AND THE RELATION WITH YOUR STUDY IS VERY REMOTE- JUST KEEP IT ECOPHYSIOLOGICAL, IT HAS A VLUE OF ITS OWN**

Response
We fundamentally disagree with the reviewer that climate change is a 'swindle' and have left the sentence in place. However, we have added the study hypothesis in this paragraph. See Line 65 -80.

Line 65 – 80: Due to the major contribution of MPB to ecosystem processes, they have been widely studied in tropical coral reefs and intertidal flats and coastal intertidal zones (Mitbavkar and Anil 2002; Mitbavkar and Anil 2004; McMinn et al., 2005; Underwood, 2002). However, there are relatively few studies of them in tropical estuaries (Patil and Anil 2008). Although the photosynthesis of tropical benthic MPB communities is likely to be highly sensitive to elevated temperature, comparatively little attention has been given to the combined effects of both elevated light and temperature. Water temperatures in tropical marine ecosystems are already high and relatively small increases can have severe negative impacts.. Unfortunately, information on the combined effects of temperature and light on the photosynthetic responses from this region is lacking. To date, the temperature tolerance of tropical microalgae has been studied mostly in plankton, which cannot usually survive temperatures above 30°C, which is consistent with the maximal temperature they normally experience (Indrayani et al., 2020; Thomas et la., 2012). MPB, however, may be exposed to much higher temperatures during low tide that can be several times higher than in the water column. Global temperatures are predicted to increase over the course of the century, and this will inevitably lead to higher temperatures in shallow benthic ecosystems. Thus, the aim of this study is to determine the stress response of MPB physiology to the combined effects of elevated irradiance and temperature and understand their subsequent recovery. We hypothesize that high light could reduce the photoprotection capacity thus causing severe damage and impair the recovery process. To this end we exposed diatom-dominated communities to a range of temperatures and used PAM fluorometry to measure photosynthetic characteristics during exposure to different light levels.

**MATERIAL AND METHODS**

**LINE 80 mangrove forest, mainly dominated by Rhizophora and Avicennia species. The benthic habitat is mostly composed of coarse sand with patches of mud. The water normally carries a high-suspended load, originating partly from the mangrove forest and from the river sediment itself. To gain a better understanding of the surrounding environment, from three sites (Sites A, B and C) all in proximity (20 meters) to the intertidal zone of the Tanjung Rhu Estuary, during both high and low tide. HIGHLIGHTED SENTENCE IS VAGUE. SHOULD BE ERASED OR EXTENDED WITH PRECISE DESCRIPTION OF HOW A SUCH A BETTER UNDERSTANDING IS PURSUED.**
**ALSO, SAMPLES FROM 2007? AN EXPLANATION FOR THIS ASYNCHRONY IS REQUIRED**

Response: Text has been modified to improve clarity. See Line 90 to 91:

Line 90 – 91: Samples were collected in April 2007 from three sites (Site A, B and C) all in proximity (20 meters) to the intertidal zone of the Tanjung Rhu Estuary, at ebb and flood tide. Description of sites A, B and C are provided in Table 1.

The samples were collected AND analysed in 2007. Furthermore, to date no further reports have made available on the photophysiology of benthic microalgae from tropical regions. Hence, data from this study is valuable.

**LINE 85 Description of sites A, B and C are provided in Table 1. Environmental data (salinity, temperature) and photosynthetic parameters were collected at ebb and flood tide. Water height at ebb tide was approximately 0 to 0.2 m and 0.5 to 1 m at flood tide. On each occasion, seven 15 mm diameter hand-pushed sediment cores were taken; three for photosynthetic parameter analysis, three for chlorophyll a analysis and one for species composition. For the temperature incubation experiments, the top 10 mm (approximately) of the sediment was manually scraped off.**

**LINE 90 and placed in a dark plastic bag (20 x 90 20 cm). Samples were stored in the dark and promptly returned (approximately 15 min) to the laboratory for the experiments. Water samples for all parameters and nutrient analyses were collected during high tide (water level: 1.0 m) and low tide (water level: 0.2 m). IT APPEARS REPEATED, HOWEVER YOU ARE REFERING TO HYDROLOGICAL VARIABLES, NOT TO WATER QUALITY WHICH INDICATES POLLUTION DEGREE**

Response:

Sentence rephased for clarity. Both environmental data and water nutrients were collected at each sampling occasions. Please see Line 97 to 104.

Line 97 – 104: At each sampling site, environmental measurements (temperature, salinity, and photosynthetically active irradiance (PAR) were measured using a Hydrolab Datasonde 4a (Hach, Loveland CO). Water samples for all parameters and nutrient analyses were collected during high tide (water level: 1.0 m) and low tide (water level: 0.2 m). Nitrate, phosphate, and ammonium were analyzed on a Hach Kit DR 2000 Spectrophotometer (Hach, Loveland CO). Ammonia, expressed as ammonia-nitrogen ($NH_3$-N), was determined using the Nessler method (Hach Company, 1995). Phosphate, expressed as phosphorus-phosphate ($PO_4^{3-}$-P) was determined by the PhosVer (ascorbic acid) method (Hach Company, 1995). Nitrate, expressed as nitrate-nitrogen ($NO_3^-$ -N) was determined by the cadmium reduction method (Hach Company, 1995).

**METHODS**
**LINE 199 STATE IF DATA WERE TESTED FOR HOMOSCEDASTICITY AND NORMAL DISTRIBUTION BEFORE APPLYING PARAMETRIC STATISTICAL TESTS, AND IF SO, WHAT TEST WAS USED**

Response:
The data was checked for normality and homogeneity of variances Shapiro-Wilk's and Levene's tests, respectively. The information has been updated in the manuscript. Please see line 188 to 190.

Line: 188 – 190: Before analyses, the data was checked for normality and homogeneity of variances Shapiro-Wilk's and Levene's tests, respectively. Data were transformed whenever necessary to comply with ANOVA assumptions. Differences were accepted as significant at $P < 0.05$ unless otherwise stated. All statistical analyses were performed using SPSS 18.0.

**RESULTS**
**ACCORDING TO WHAT IT IS READ IN (METHOD) ON COMMUNITY COMPOSITION (LINES 100-110), WHAT IT IS PRESENTED AS RESULT:**

**(LINE 195) The diatom genera Cocconeis, Navicula and Rhopalodia dominated the benthic habitats, which predominantly consisted of course sand with mud/silt patches. The sediment characteristics were similar among the sampling sites. The most common species found were Cocconeis placentula, Navicula raphoneis, Navicula clipeiformis and Rhapalodia acuminatea. The composition of the diatom communities at all sites and tides were similar without any differences in dominant species.**
**IS NOT BASED ON PRESENTED EVIDENCE, SUCH AS SPECIES LIST, IMAGES, RELATIVE ABUNDANCES SIMILARITY MEASUREMENTS**
**HENCE, IT WOULD BE ADEQUATE TO REMOVE ANY PARTICULARITY ON COMMUNITY ANALYSIS, BOTH IN METHOD AND RESULTS I SUGGEST TO FURTHER WORK SAID DATA ON DIATOMS AN TRY TO PUBLISH INDEPENDENTLY**

Response
We have changed the manuscript according to the suggestion and hence, the method and results on community analysis were removed, and only the common species are noted.
Method: Section 2.2 Community composition has been removed.
Results: Rephased accordingly. Please see Line 192.
Line 192: The diatom genera *Cocconeis, Navicula* and *Rhopalodia* (observed by the author) dominated the MPB communities, which predominantly located on coarse sand with mud/silt patches.

**LINE 206 The highest nutrient levels were recorded during high tide, when seawater re-entered the estuary (Table 2 and Table 4(a)) There was a least a 50% increase in nitrate and phosphate levels during high tide, although the changes were not significant (Table 2).**

**MAYBE BECAUSE THE STATISTICAL TESTS USED (PARAMETRIC) WERE NOT APPROPIATE?**

Response: We acknowledge an error on reporting the data. A 50 % increase were observed in Phosphate and significant difference (p < 0.001) was observed between tides. Sentence rephased for clarity. Please see Line 200 to 203.

Line 200 – 203: The highest nutrient levels were recorded during high tide, when seawater re-entered the estuary (Table 2 and Table 4(a)) There was a least a 50% increase in nitrate and phosphate levels during high tide, although the changes were not significant (Table 2).

**DISCUSSION**
**LINES 240-258 DO NOT PRESENT ANY INFERENTIAL CONTRIBUTION (THEORY); MOST LIKELY BECAUSE NO WORK WAS DONE ON A MOST NEEDED HYPOTHESIS**

**LIKEWISE**
**LINES 260-285 WOULD DO BETTER CONTRASTING A HYPOTHESIS AND NOT JUST COMPARING WITH OTHER STUDIES, WHICH HAS TO BE DONE WHEN STATING THE PROBLEM/HYPOTHESIS**

Response: This paragraph was intended to discuss the in situ environment of Tanjung Rhu estuary, emphasising on the impact of high light and temperature on benthic diatom. However, the paragraph was rephased for clarity. See Line 235 to 255.

Line 235 – 255
In the present study, we found that the MPB communities inhabiting the Tanjung Rhu estuary were not inhibited by the high and variable light environment (range: 170 μmol photons m$^{-2}$ s$^{-1}$ - 1900 μmol photons m$^{-2}$ s$^{-1}$), or temperatures (28.5 °C to 43 °C) during tidal exposure. The surface water temperature increased from 28.5 °C to 43 °C during high to low tide. The emersion at low tide caused the diatoms to migrate into the sediment to avoid excess light as a photoprotective mechanism. During the tidal cycle, the MPB communities were able to acclimatize to these changes by optimizing photosynthesis while minimizing photodamage caused by temperature and light fluctuations through physiological changes (e.g., diversion of excess energy away from photosystem reaction centres) (Consalvey et al. 2005, Coelho et al. 2011, Serôdio et al., 2005).

Maximum quantum yield ($F_v/F_m$) values are often used as a sensitive indicator of photosynthetic stress (Du et al., 2018; McMinn et al., 2005). In a review by Campbell and Tyystjärvi (2012) it was suggested that for photoinhibition measurements dark-adapted $F_v/F_m$ values can be used if measuring the rate of oxygen evolution is not possible. Thus, in this study, dark-adapted $F_v/F_m$ values were used as a proxy of diatom health/stress (Consalvey et al., 2005; Du et al., 2018) upon exposure to light and temperature stress. An increase in temperature and irradiance during low tide caused a decline in $F_v/F_m$ at all sites in the Tanjung Rhu estuary; these were low compared with optimum values of ~ 0.650 for healthy microalgae (McMinn and Hegseth, 2004). Similarly, low values were found by McMinn et al. (2005) on the tropical shore of Muka Head and Songsong Island, Malaysia, where the average maximum quantum yields, which were only 0.325, were probably caused by either high irradiances or nutrient stress. Falkowski and LaRoche (1991) suggested that nutrient limitation would decrease the optimal quantum efficiency of phytoplankton. However, mangrove ecosystems are highly dynamic, and support relatively high microalgae biomass and nutrient availability should not be a limiting factor (Hilaluddin et al., 2020; Rahaman et al., 2013). Nutrient concentrations (phosphate and nitrate) measured in this study were within the average range for other tropical mangrove estuaries (Rajesh et al. 2001) and are unlikely to be limiting. Thus, it is likely that the high temperatures, of up to 42.5 °C during low tide in this study, were contributing to the low $F_v/F_m$ values.

**LINES 286-389 I FEEL ARE ON THE RIGHT TRACK IN TERMS OF ECOPHYSIOLOGICAL STRATEGIES THAT EXPLAIN THE PERMANENCE AND ABIDENCE OF DIATOM POPULATIONS UNDER THESE CONDITIONS**

Response: Thank you

**LINE 390 DIATOMS ARE NOT PLANTS!**

Response: Replaced 'plant' with protist. Line 364

**CONCLUSION**
**I FIND CONCLUSION WELL DERIVED, BUT IN AGREEMENT WITH THE ECOPHYSIOLOGICAL STRATEGY VIEW OF DIATOMS, AND NOT IN THE TERMS THAT I HAVE OBSERVED WHICH SHOULD BE AVOIDED GLOBAL WARMING OR BYPASSING THE FACT THAT DIATOM POPULATIONS ARE WELL ADAPTED TO EXTREME CONDITIONS, WHETHER TROPICAL AS THESE OR IN DESERT SABKAHS.**

**HOWEVER, THE STARTING SENTENCE SHOULD BE ELIMINATED:**

**LINE 405 While the data presented herein demonstrates the response of tropical benthic diatoms to high irradiance and extreme temperature, the use of RLC data alone prevents a more rigorous examination of non-photosynthetic quenching activities such as the xanthophyll cycle (Cartaxana et al., 2013).**

Response:

Line 405 deleted as per suggestion.

---

## Referee Report (RR1)

Aninda Mazumdar
**Associate Editor to Biogeosciences**
Title: **Tolerance of tropical marine microphytobenthos to elevated irradiance and temperature**
Author(s): Sazlina Salleh and Andrew McMinn
MS No.: bg-2021-18
MS type: Research article
Iteration: Initial Submission

DEAR ANINDA, PLEASE FIND IN THE FOLLOWING MY REITERATIVE CRITIC AND RECOMENDATION ON THE ASIGNED PAPER.
MY OPINION IS THAT: THIS MANUSCRIPT STILL HAS GOOD POTENTIAL IN TERMS OF ECOPHYSIOLOGY AND SHOULD BE EVENTUALLY PUBLISHED. HOWEVER, IT HAS TO BE FURTHER MODIFIED ACCORDING TO THE REITERATED OBSERVATIONS .

TITLE: Response of tropical marine benthic diatoms exposed to elevated irradiance and temperature
Title changed as suggested. New Title: Tolerance of tropical marine microphytobenthos to elevated irradiance and temperature

AUTHORS HAVE COMPLIED WITH MOST CORRECTIONS AND SUGGESTIONS INDICATED BY THE OTHER REVIEWER AND I. HOWEVER, IT IS MY APPRECIATION THAT THE FOLLOWING REMARKS HAVE NOT BEEN TAKEN SERIOUSLY, EVEN THOUGH SAID ISSUE CONSTITUTES AN ESSENTIAL BASIS FOR THE SCIENTIFIC SOUNDNESS OF THIS MANUSCRIPT; THEY WERE NOT SUGGESTIONS.

FIRST NOTICE
LINE 12 Abstract. Shallow tropical marine environments are likely to experience future water temperatures that will challenge the ability of life to survive. Here, the response of a Malaysian (IRRELEVANT) microphytobenthic community to temperature and light was examined.
FUTURE RISE IN WATER TEMPERATURE? HOW FAR INTO THE FUTURE....15 YEARS (NOW)? THIS PREMISE LACKS SCIENTIFIC OR PHILOSOPHICAL (LOGIC) BASIS

Response:
We disagree that references to climate change should be removed from this manuscript. Climate change provides the underlying reason for undertaking this research. Examining the effects of elevated temperatures, such as those predicted to occur this century, is a completely reasonable and legitimate approach. Thank you.
UNQUESTIONED ACCEPTANCE OF A THEORY RENDERS IT QUESTIONABLY UNSCIENTIFIC BUT, INASMUCH THIS IS LINKED TO THE PROBLEM POSED IN INTRODUCTION….SEE BELOW

LINE 75 Future warming is likely to cause this temperature to occur more frequently, which will cause a reduction in benthic primary production
TO MUCH CONFIDENCE ON THE GLOBAL HYPERWARMING EXPECTATIONS MAKE THIS CONCLUDING REMARK OUT OF PLACE (DISCRETE DATA CONTRADICTING ECOLOGICAL PLANETARY EVOLUTION) BESIDES, BY THE TIME SUCH TEMPERATURE RISE WOULD MANIFEST (IF EVER) A REPLACEMENT BY OTHER (PRIMARY PRODUCERS) SPECIES OF THE MFB WOULD HAVE OCCURRED COMPENSATING THE ALEDGED REDUCTION… NATURE WORKS IN SUCH A WAY!

Response:
We disagree with the reviewer about the consequences of global warming and intend to leave these sentences in. Thank you.

AS I STATED AT THE START OF THIS REVIEW, THIS IS A MATTER OF UTMOST RELEVANCE. AUTHORS SAY THAT "We disagree that references to climate change should be removed from this manuscript." BUT THEY PROVIDE NO REFERENCES IN THE EDITORIAL SENSE! I.E., BADLY NEEDED CITATIONS. OTHERWISE, THE SOLE BACKGROUND MAY BE TRACED TO THE INITIAL PROPAGANDA DRIVEN STATEMENTS MADE BY MARGARET TATCHER, AL GORE, AND LATER BY LEONARDO DI CAPRIO, WHICH HAVE DUBIOUS AGENDAS, OR THE POLITICAL REPORT BY THE IPCC.SCIENTIFIC PAPERS THAT SOLIDLY SUPPORT THE EXPECTED RISE IN TEMPERATURE, WITHOUT AN HOMEOSTATIC RESPONSE BY THE PLANETARY ECOSYSTEM, SHOULD BE PROVIDED AND, MAINLY, DISCUSSED…

THE STATED DISAGREEMENT BY THE AUTHORS WITHOUT EPISTEMOLOGICAL BASED ARGUMENTS ARE NOT ACCEPTABLE.

I DO NOT INTENT TO UNDERMINE THE VALUE OF THIS STUDY BUT TO UNDERLINE THE RELEVANCE IT MAY HAVE IF THE ABOVE REMARKS ARE DULY CONFRONTED

MINOR OBSERVATIONS

IN ALL TABLES, SIGNIFICANCE LEVELS (ALPHA VALUES) SHOULD BE REFERED CONSISTENTLY AS 0.05, OR EXPLAIN WHY A HIGHER CONFIDENCE (V.GR.,0.001) WAS SET

REPLACE DIATOMS FOR MPB

Table 3: In-situ photosynthetic parameters of benthic diatom at Tanjung Rhu estuary (Site A, B and C) in the surface 10mm (depth ~ 0.2 m at low tide and 1.0 m at high tide). Data are means ± SD, n = 3.

Table 5 : Two way ANOVA of effects of temperature (0 o C to 60 o C) and irradiance level (1800, 890 and 0 µmol photons m -2 s -1 ) on the photosynthetic parameters (F v /F m , maximum quantum yield; α, photosynthetic efficiency; rETRmax, relative electron transport rate; E k , photoacclimation index), NPQ (non-photochemical quenching) and recovery rate the benthic diatom after exposure to temperatures treatment for 1 hour.

OTHER
Fig. 4: Recovery rate s -1 of F v /F m after the RLC for low irradiance experiments. Recovery rates were plotted against experimental temperatures at irradiances of (1800 µmol photons m -2 s -1 (closed square), 890 µmol photons m -2 s -1 (closed circle) and 0 photons m -2 s -1 (open triangles)) and temperatures (30, 35, 40, 45, 50, 55 and 60 o C). In this recovery analysis, samples of 30 o C and 35 o C were analyzed without replicates; thus, no error bar was obtained.

---

## Author Response (AR2)

**RESPONSES TO REVIEWER'S COMMENTS**

Manuscript ID: bg-2021-18

Dear Editor,

Thank you for giving us the opportunity to submit a revised draft of the manuscript *"Tolerance of tropical marine microphytobenthos exposed to elevated irradiance and temperature"* for publication. We have incorporated the comments made by the reviewer. Those changes are highlighted within the manuscript. Please see below, in yellow, for a point-by-point response to the reviewer's comments and concerns.

**Comments**

**Title:**

Title changed: Tolerance of tropical marine microphytobenthos exposed to elevated irradiance and temperature

**Abstract**
**FIRST NOTICE**
LINE 12 Abstract. Shallow tropical marine environments are likely to experience
future water temperatures that will challenge the ability of life to survive. Here,
the response of a Malaysian (IRRELEVANT) microphytobenthic community to temperature
and light was examined.
FUTURE RISE IN WATER TEMPERATURE? HOW FAR INTO THE FUTURE....15 YEARS
(NOW)?
THIS PREMISE LACKS SCIENTIFIC OR PHILOSOPHICAL (LOGIC) BASIS

Response: Statement on climate change/future warning was removed and rephased accordingly.
See Line 12 -14 Page 1
"Malaysia" removed from the sentence.

LINE 75 Future warming is likely to cause this temperature to occur more frequently, which will cause a reduction in benthic primary production
TO MUCH CONFIDENCE ON THE GLOBAL HYPERWARMING EXPECTATIONS MAKE THIS CONCLUDINGREMARK OUT OF PLACE (DISCRETE DATA CONTRADICTING ECOLOGICAL PLANETARY EVOLUTION) BESIDES, BY THE TIME SUCH TEMPERATURE RISE WOULD MANIFEST (IF EVER) A REPLACEMENT BY OTHER (PRIMARY PRODUCERS) SPECIES OF THE MFB WOULD HAVE OCCURRED COMPENSATING THE ALEDGED REDUCTION… NATURE WORKS IN SUCH A WAY!

Response: Statement on climate change/future warming was removed from the text and rephased. See line 76 – 80 Page 3 and Line 378 Page 12

IN ALL TABLES, SIGNIFICANCE LEVELS (ALPHA VALUES) SHOULD BE REFERED CONSISTENTLY AS 0.05, OR EXPLAIN WHY A HIGHER CONFIDENCE (V.GR.,0.001) WAS SET.

Response: Thank you for the comment. We have made the changes throughout the manuscript.

REPLACE DIATOMS FOR MPB

Response: Thank you for the comment. We have made the changes throughout the manuscript.

Other

Fig. 4: Recovery rate s$^{-1}$ of Fv /Fm after the RLC for low irradiance experiments. Recovery rates were plotted against experimental temperatures at irradiances of (1800 μmol photons m$^{-2}$ s$^{-1}$ (closed square), 890 μmol photons m$^{-2}$ s$^{-1}$ (closed circle) and 0 photons μm$^{-2}$ s$^{-1}$ (open triangles)) and temperatures (30, 35, 40, 45, 50, 55 and 60°C). In this recovery analysis, samples of 30°C and 35°C were analyzed without replicates; thus, no error bar was obtained.

Response: Thank you for pointing the error. Corrections have been made. See Figure 2 Page 24, Figure 3 and 4 Page 25